# Blockchain Processing Technique Based on Multiple Hash Chains for Minimizing Integrity Errors of IoT Data in Cloud Environments

**DOI:** 10.3390/s21144679

**Published:** 2021-07-08

**Authors:** Yoon-Su Jeong

**Affiliations:** Department of Information and Communication Convergence Engineering, Daejeon-si 35349, Korea; bukmunro@mokwon.ac.kr; Tel.: +82-42-829-7678

**Keywords:** IoT, data integrity, hash chain, data error process, load balance

## Abstract

As IoT (Internet of Things) devices are diversified in the fields of use (manufacturing, health, medical, energy, home, automobile, transportation, etc.), it is becoming important to analyze and process data sent and received from IoT devices connected to the Internet. Data collected from IoT devices is highly dependent on secure storage in databases located in cloud environments. However, storing directly in a database located in a cloud environment makes it not only difficult to directly control IoT data, but also does not guarantee the integrity of IoT data due to a number of hazards (error and error handling, security attacks, etc.) that can arise from natural disasters and management neglect. In this paper, we propose an optimized hash processing technique that enables hierarchical distributed processing with an n-bit-size blockchain to minimize the loss of data generated from IoT devices deployed in distributed cloud environments. The proposed technique minimizes IoT data integrity errors as well as strengthening the role of intermediate media acting as gateways by interactively authenticating blockchains of n bits into n + 1 and n − 1 layers to normally validate IoT data sent and received from IoT data integrity errors. In particular, the proposed technique ensures the reliability of IoT information by validating hash values of IoT data in the process of storing index information of IoT data distributed in different locations in a blockchain in order to maintain the integrity of the data. Furthermore, the proposed technique ensures the linkage of IoT data by allowing minimal errors in the collected IoT data while simultaneously grouping their linkage information, thus optimizing the load balance after hash processing. In performance evaluation, the proposed technique reduced IoT data processing time by an average of 2.54 times. Blockchain generation time improved on average by 17.3% when linking IoT data. The asymmetric storage efficiency of IoT data according to hash code length is improved by 6.9% on average over existing techniques. Asymmetric storage speed according to the hash code length of the IoT data block was shown to be 10.3% faster on average than existing techniques. Integrity accuracy of IoT data is improved by 18.3% on average over existing techniques.

## 1. Introduction

Over the past few years, the IoT (Internet of Things) has been utilized in a variety of areas such as healthcare, environment, military, transportation, and IT [1], as various technologies related to the IoT have emerged in the cloud environment [1]. However, due to the limited computing resources of IoT, various studies are underway to solve various problems related to IoT data processing (network traffic, data processing delay, data uncertainty, data integrity, etc.).

Cloud and IoT computing environments should be able to ensure large-scale IoT network delay and data protection, but current operating IoT computing environments require new solutions that reflect IoT characteristics [2] because information is processed in a cloud-centric way. In particular, IoT computing environments should be able to minimize latency by pre-processing information control and accessibility in IoT computing environments by granting different access policies between IoT networks through pre-analysis and prediction of information collected from IoT devices. In existing operational cloud computing environments, some computationally intensive tasks should be offloaded to remote cloud servers to increase the accuracy of the information sent and received.

Recent studies have focused on generating blockchain to enable service and smart contract sharing [3,4,5]. In particular, blockchain used in IoT systems is not only used to build trust in IoT systems, but it is also used in applications as a way to boost the IoT sharing economy [6,7,8,9,10]. Off-chain transactions are also used to reduce the overhead of trackjacks so that economic exchanges between IoT components can be made using IoT services. Methods to reduce transaction overhead using off-chain transactions present a simple transaction scenario for IoT systems [11,12]. In areas where low-cost devices are used, lightweight schedulers are also being studied to split DNN operations between IoT devices and servers (data centers) to solve latency and integrity problems [13,14]. Furthermore, to operate the offload of hierarchical deep learning tasks, we treat CNN models separately as upper and lower layers. However, these methods have the problem of not activating the entire layer of DNN.

In this paper, we propose an optimized hash processing technique that can minimize the integrity errors of data generated by numerous IoT devices deployed in distributed cloud environments. The proposed technique guarantees the integrity of IoT data by hashing IoT data from servers (data centers) adjacent to IoT devices to a blockchain instead of cloud servers. Specifically, the proposed technique hierarchically distributes the two-way authentication of n bits of IoT information by grouping index information of IoT data distributed at different locations into the blockchain to maintain the integrity of the data. The reason is to ensure the reliability of IoT information by verifying the hash value of IoT data in the process of storing data collected from IoT devices into a blockchain. The proposed technique used polynomial multiplication and security comparison to allow IoT data to be configured as building blocks to be optimized for distributed environments, and load balance was performed after asymmetric hash of IoT data to minimize errors in IoT data integrity. Through this process, the proposed technique allows minimum errors of collected IoT data while ensuring the linkage of IoT data.

The reminder of this paper is organized as follows: Section 2 provides the integrity-related studies of data processed in a cloud environment. Section 3 proposes optimization techniques to minimize integrity errors in IoT data. In Section 4, we evaluate the performance of the proposed technique, and we conclude in Section 5.

## 2. Related Works

As various services are provided in the cloud environment, the requirement to maintain the integrity and security of service-specific data is increasing. However, blockchain-related application research has recently increased as various services are used in cloud environments, causing problems such as data corruption and deletion.

V. Yatskiv et al. proposed a convergence technology between blockchain technology and cloud services to protect unauthorized changes to video files [15]. This technique allows each frame in a video file to be computed as a hash function and to form a block sequence of the computed hash function for all frames. In addition, this technique transfers all hash block information to blockchain cloud storage so that hash blocks in all video frames can be trusted by all stakeholders. However, since this technique assumes the availability of the entire video to obtain the hash, the integrity depends on the reliability of each frame when it leaves the IoT device.

P. Gallo et al. proposed a blockchain-based video surveillance system to protect camera settings, locations, and directions and provide video flow integrity [16]. The system jointly provided verification and invariance of the video so that video frames, timestamps, and camera settings could not be digitally controlled by malicious users, making them available only to authorized users in the event of an incident. However, the system poses a problem of availability and privacy conservation when heterogeneous sources are interrelated.

A. Knisch et al. describes verifiable privacy preservation using session proof consistency algorithms and image protection systems based on the Ethereum blockchain [17]. This technique provides evidence for compression or low-quality copies of images so that rigorous proof of ownership is feasible only for dedicated private Ethereum blockchains. However, this technique is effective only when smart contracts are used only in private blockchains or transferred to other blockchains optimized for storage.

G. Dong et al. proposed a secure IoT data integrity audit scheme that provides anonymity, fairness, accountability, security, etc. based on the Hyperledger Fabric [18]. The core content of the IoT data integrity audit framework is to utilize open and non-detective information of blockchain. Furthermore, the technique leverages the characteristics of smart contract auto-execution and thus provides fairness efficiently. However, this technique requires further experiments on anonymity, fairness, accountability, security, etc. to be applicable in the real world.

E. A. Kanimozhi et al. applied blockchain technology when storing data on cloud servers, proposing a verification scheme that can identify data corruption at the time of fraud if cloud authorities collide with third-party validators [19]. The technique incorporates invariance and variance among the properties of the blockchain to identify unauthorized changes in the data, as well as enable data recovery due to local copies of the blockchain. However, this technique does not fully address the problem of data recovery because it does not have protective properties for various data.

M. Ramkumar proposed a framework to ensure the integrity of blockchain-based information system processes [20]. This framework guarantees the integrity of information systems in blockchain networks by running finite state machine models. In particular, the framework performs comparisons and contrasts with existing blockchain-based frameworks such as the Clark–Wilson (CW) system’s integrity model and Ethereum. However, the framework requires a simple design of byte codes in a blockchain-based system integration (BCSI) to extensively investigate nested Merklet tree (OMT) structures.

M. K. Choi et al. proposed a system that uses blockchain technology to monitor the integrity of the data in programmable logic controllers (PLCs) [21]. The system uses the proof of monitoring (PoM) concept for data integrity in PLC to solve cybersecurity in nuclear power plants. Furthermore, the system used false data injection attacks to verify the safety of the system. However, it is necessary to verify and evaluate that the safety system does not affect the performance of the safety system to apply it to real-world plant systems.

A. Majumdar et al. proposed a technique that combines with a robust exchange proof protocol called proof of integrity without cryptocurrency or other domain-specific elements [22]. This technique provides distributed data security and operational continuity while generating a trusted layer for application-specific specificity of domain-specific elements. However, this technique is not easy to apply to large networks without specific data loss because it uses XOR-like operations to determine the existence and absence of data.

D. Yue et al. proposed a blockchain-based framework to verify the integrity of the data stored on P2P cloud storage [23]. The framework validates data integrity using the Merkle tree and presents a sampling strategy to better perform sampling verification. However, since this framework has not evaluated its performance with real-world blockchain systems, it is necessary to improve the equations for verifying the integrity of the data.

B. Ravishankar et al. describes the requirements of cloud environments and the adoption of blockchain-based databases about data integrity [24]. In particular, the database performs a blockchain-based preliminary design of the database to process the data effectively. However, while this technique can secure the integrity and simplify thread addressing among distributed replicated copies, it needs to be supplemented because one violation of a single miner can seriously affect the overall miner availability.

P. Huang et al. proposed a collaborative audit blockchain framework to address trust issues between data owners and cloud service providers [25]. The framework uses an open audit method in which third-party auditors are delegated to interact with cloud service providers for traditional auditing operations. In addition, the framework can replace third-party auditors because all consensus nodes are executing audit delegations, preventing entities from deceiving each other. However, this framework requires the introduction of public auditing methods in which third-party auditors are delegated to interact with cloud service providers for auditing tasks. Furthermore, it is difficult for centralized third parties to remain neutral, and there is a problem in which remote data audits are exposed to some threats, such as collusion attacks.

## 3. Optimization Techniques to Minimize Integrity Errors in Internet of Things (IoT) Data

Over the past few years, IoT devices have been used in various fields, including health care, the environment, transportation, telecommunication, and military, and IoT devices have been trying to create new businesses through the development of applications that can be used in the cloud environments. However, IoT devices do not guarantee the integrity of IoT data due to various technical problems such as network bandwidth, data processing time, and security in the process of delivering IoT information to the server. In this paper, we propose an optimized hash processing technique that can minimize the integrity errors of data generated by numerous IoT devices deployed in distributed cloud environments. The proposed technique aims to perform two-way authentication and distributed processing by separating the index information of IoT data located at different layers by n-bit size for integrity verification of IoT data bound by blockchain, ensuring the reliability of IoT data.

### 3.1. Overview

Until recently, IoT-related research has centered on studies that ensure the integrity of IoT data by safely processing IoT data transmitted and received in distributed cloud environments without errors. In particular, blockchain-based IoT data processing studies are continuously underway to minimize network delays in cloud environments, which can minimize bottlenecks in overlay networks and latency in processing IoT data.

In this work, we propose an optimized hash processing technique that enables hierarchical distributed processing with an n-bit size blockchain to minimize the loss of data generated from IoT devices deployed in distributed cloud environments. The proposed technique can minimize IoT data integrity errors by strengthening the role of intermediate media acting as gateways and reducing the load on cloud servers by authenticating the blockchain of n bits to the n + 1 layer and n − 1 layer to verify IoT data normally in low latency. To minimize IoT data integrity errors, the proposed technique has the following characteristics.

First, the proposed technique groups encrypted IoT data into subnets so that encrypted IoT data is stored in distributed sources using blockchain to minimize the integrity error of IoT data.

Second, IoT data processed in the proposed technique consist of building blocks through polynomial multiplication and security comparisons, ensuring parameters of IoT data without illegal intervention by third parties.

Third, after asymmetric hash processing of IoT data, IoT data are optimized through load balance to secure the linkages of IoT data as well as the speed of sending and receiving IoT data.

As shown in Figure 1, the proposed technique can be directly taken in real-time to minimize the loss of integrity of IoT data as well as bottlenecks in overlay networks by hashing IoT information without separate behavior of IoT devices using n -bit blockchain information. The proposed technique constructs IoT data to interleave any k bits of IoT data by extracting k information of IoT properties ($pi_{k}\in$ PI (1 ≤ k ≤ n)) for the operational process in Figure 1. The reason for doing this is to check the integrity of IoT data immediately without additional action to act on anomaly symptoms. Furthermore, verifying the hash information value of the blockchain of IoT information distributed in distributed cloud environments is to ensure the reliability of IoT data.

### 3.2. System Architecture

The proposed technique constructs a system model (IoT device layer, cloud edge layer, load balance layer, preprocessing layer, cloud layer, etc.) as shown in Figure 2 to efficiently verify the integrity of IoT data in distributed cloud environments.

#### 3.2.1. Device Layer

The device layer is responsible for transferring information from numerous IoT devices distributed in distributed cloud environments to cloud edge devices. This layer is natively supported over the RTP protocol for different types of information that IoT devices can collect. The connection of the server is checked through an Internet communication channel.

#### 3.2.2. Cloud Edge Tier

The cloud edge layer plays a role in reducing network overhead through bandwidth savings and type-specific classification of information on IoT data before forwarding data collected from IoT devices to higher layers.

#### 3.2.3. Load Balance Layer

The load balance layer negotiates with the multi-hash function h: ${\left\{0,1\right\}}^{*}$ → ${\left\{0,1\right\}}^{k}$ before sending IoT data types from the cloud edge layer to the higher layer. The reason for generating arbitrary block $b_{i}$ with n bits is to simply categorize the complex information sent and received from IoT devices.

#### 3.2.4. Treatment Layer

The processing layer further enforces the load balance of IoT data after detailed classification of IoT data classified as pre-processed n-bit blocks in the load balance layer by different types of information. This process can maintain the integrity and accuracy of large amounts of IoT data because it can directly access IoT data. Also, this process requires more computational power than the load balance layer but has the advantage of saving bandwidth and maintaining the recognition results of large amounts of IoT data.

#### 3.2.5. Cloud Layer

The cloud layer stores IoT data delivered from the processing layer in a database and performs error checking by different types of IoT data. The cloud layer determines whether or not to provide services through an error check process. When the service is determined, it is responsible for creating keys for service authentication and managing the generated items.

### 3.3. Creation of IoT Information

To generate IoT information, the proposed technique first generates blockchain-based data generated from all IoT devices distributed in distributed cloud environments. In the proposed technique, IoT data are generated as information blocks in 64-bit units, and to improve the throughput of cloud servers, block information in 64-bit units is preprocessed as a correlation matrix, such as Equation (1).
(1)$$v_{i}=\;\mathrm{Map}:\left(id_{i},block_{i}\right)$$

Pre-processed IoT data, such as Equation (1), allow different types of information-type IoT data to be processed differently depending on the type of service by generating blocks in 64-bit units, such as Equation (2).
(2)$$V_{n}=v_{i}\in \;\{\;Z\;|\;v_{1},\;v_{2},\;\cdots ,v_{n}\}$$
where $v_{i}$ means the i-th block. Z is used to sequence each IoT information type when processing different types of IoT information differently as information.

The block $V_{n}$ generated by Equation (2) consists of two hash chains depending on the number of replications of od/ven in IoT data. This process has the advantage of being able to process continuous hashes without loss of information in adjacent IoT data. Furthermore, it has features that allow it to optimize IoT data in distributed cloud environments, rather than just having to lower the overhead than verifying the integrity of IoT data through signatures.

### 3.4. Properties Creation by IoT Data Type

The proposed technique generates properties by type of IoT information, weighting the block of IoT data in blockchain units to check the verification of IoT data, extracting the property values of IoT block data as shown in Equation (3).
(3)$$P_{i}=p_{i}\in \;\{\;V_{i}\;|\;1\leq \;i\;\leq \;n\;\mathrm{mod}\;64\}$$
where $V_{i}$ means the property value for the i-th block in 64-bit units, and $p_{i}$ means the IoT information property in 64-bit units.

Equation (3) combines only the attribute values of IoT data among different types of information-type IoT data into a hash function and treats several attribute values as a group of 64-bit units. Equation (3) is used with the value of the hash chain configured by odd/ven, and the information in Equation (3) is used as a function to verify the integrity of IoT information by adding it to the first and last of the hash chain.

Figure 3 shows the process of grouping multiple hash chains by generating properties of IoT data in 64-bit units. In Figure 3, IoT data select seed $\hat{S_{t}}$ based on IoT block information’s similar information. At this time, IoT data are grouped in a highly relevant order of IoT data blocks. The proposed technique will create a pattern of IoT information blocks $b_{t}$ by applying a polynomial of N − 1 order when the entire path of the hash chain is grouped.

### 3.5. Block Processing for Hash Processing of IoT Data

When delivering information collected from IoT devices to cloud servers, it selects the optimal scheduler policy for IoT data in the process of collecting through the cloud scheduler process according to the cloud scheduler policy.

The proposed technique will manage data collected from different IoT devices to cloud edge devices. The proposed technique verifies that the cloud edge device exists in the closest location from the IoT device when selecting it. This reason is to minimize IoT data processing latency. Furthermore, the proposed technique also treats the verification processing time due to IoT data processing differently due to cloud network delay, so IoT devices and cloud edge devices exist in the nearest network location and cloud servers, and many load balance devices are hierarchically distributed.

The proposed technique allows servers (or data centers) that receive IoT data to check whether or not IoT data is changed, and the change information is informed to IoT and the IoT device to process the change information immediately.

The proposed technique hashes IoT data generated by IoT devices to reliably process IoT data blocks in multi-cloud environments through the process shown in Figure 4.

As shown in Figure 4, the proposed technique allows IoT data blocks contained in lower groups to be organized into hierarchical multi-step groups to be periodically reconstructed according to the nature of IoT block data to tie IoT block information hierarchically by replication.

### 3.6. IoT Data Block Processing

The proposed technique delivers IoT data to multi-hash chain-based blockchains for transactions of IoT data to link different types of IoT data. The proposed technique periodically updates the connection probability of IoT data according to the probability value of the hash-chain of IoT data according to characteristics of IoT data, as shown in Figure 5. In Figure 5, IoT data are collected at regular time intervals depending on the speed, presence and absence of the nodes. The scenarios used in Figure 5 are distinguished by the speed of movement of the nodes and their presence and absence. Scenario A refers to an environment that travels at high speed (more than 30 km/h), Scenario B refers to an environment that collects IoT information in a fixed place such as an industrial IoT, and Scenario C refers to an environment that collects information on IoT devices that travel at low speed (less than 30 km/h).

The proposed technique selects seeds among the collected information and then correlates the index values of the selected information with the IoT data block. Through this process, the proposed technique minimizes bottlenecks by collecting and linking the collected information with time-series information to ensure the fast processing and integrity of IoT data.

The proposed technique groups n block information $x_{1}$, $x_{2}$, $x_{3}$, ⋯, $x_{n}$ so that they are orthogonal to each other to process the linked information of IoT data collected from IoT devices. In the proposed technique, we perform a group of linked information processes of IoT data that have ended the load balance process as shown in Equation (4).
(4)$$g≅c_{1}x_{1}+c_{2}x_{2}\;+\;\cdots \;+c_{n}x_{n}$$

A group of linked information in Equation (4) applies a probability function such as Equation (5) to use the importance of linked information as a probability condition.
(5)$$E(B_{i}^{x})=-\sum_{i=1}^{n}\frac{1}{n}log\frac{1}{n}\;=\log\;n$$

The proposed technique treats the correlation information between block information collected from IoT devices as a matrix, so it is hierarchically tied according to the weighted information of IoT data and determined by the strength of the threshold limit based on results such as high access paths or regular expression filtering checks. The proposed technique utilizes load-balancing information processed by the hash function for the linkage information of IoT data, thereby reducing bandwidth and eliminating redundant IoT data, thus maintaining information linkage between IoT data to ensure integrity.

## 4. Evaluation

This section is based on Monte Carlo, which parameters are used in simulation to verify the integrity of IoT data deployed in distributed cloud environments. Performance evaluation evaluated IoT integrity processing time, blockchain creation time when linking IoT information, asymmetric storage speed of IoT data according to hash code length, and IoT data storage efficiency of distributed cloud servers. In simulations, IoT devices that make up distributed networks exploit IoT Arduino equipment. IoT devices randomly generate packets and send them to other IoT devices, and each AP’s base station consists of a multi-mesh network. Each IoT device that makes up the network is identified by an IoT recognizer. Furthermore, each base station allows the server to collect and analyze IoT information transmitted and received from the antenna in real-time. At this time, we assume that all IoT information packets are passed on to the relative IoT device in time and that all IoT devices can be synchronized.

### 4.1. Environment Setting

The proposed techniques set the simulation environment as shown in Table 1. The proposed technique used the deep learning toolbox MatConvNet [26].

The proposed technique assumes that IoT devices were transmitted and received IoT information using IoT Arduino equipment such as Figure 6 and that each AP’s base station consists of a multi-mesh network.

The simulation uses Ubuntu 16.04.6 LTS server with Intel Xeon CPU E5-2630 @ 2.1 GHz, 128 GB memory, and 2080 Ti GPU, and parameters used in simulation environments are set based on Monte Carlo, such as Table 2.

The parameters used in the simulation environment are set based on Monte Carlo, as in Table 2. The network range is 150 m and the bandwidth is 10 MHz/5 MHz. The average transaction was set to 100–200 B and the threshold time for the variable compensation coefficient was set to 0.001 s/KB. Other parameters are shown in Table 2.

In the proposed technique, IoT data from servers adjacent to IoT devices are hashed into the blockchain rather than cloud servers to ensure the integrity of the IoT data. Each IoT device participating in the blockchain operates according to the generated events (e.g., block generation and message exchange). Each event uses parameters such as block size, number of nodes, number of adjacent nodes of all nodes, block generation capacity of all nodes, network bandwidth between each pair of regions (up/down), and average network delay between each pair of regions. Each block is generated by probability conditions according to the importance of linked information and propagates along with blockchain networks, evaluating data processing time, data storage efficiency, and IoT data accuracy.

### 4.2. Performance Analysis

#### 4.2.1. Evaluation of IoT Data Processing Time

IoT data processing time is evaluated using the number of processed transactions per second that match the hash value among all transactions entered into the server (data center). Data processing time utilized the total number of blocks entered into the server and the number of blocks processed per second. Table 3 shows the results of evaluating the number of transactions processed per second on IoT devices and servers (data centers). To produce the results of Table 3, the IoT device of the proposed technique is implemented with Raspberry Pi 3, and then the number of transactions processed per second is set to be processed according to the number of IoT devices. The results of Table 3 show that the proposed technique reduces average processing time by 2.54 times compared to the methods handled in traditional distributed cloud environments. These results are because those IoT data collected from IoT devices have been linked to IoT data by constructing IoT data into building blocks through polynomial multiplication and security comparisons in the process of encrypting them into the blockchain. Furthermore, it is a result of the merging of IoT data by clustering and based on blockchain and probability theory.

#### 4.2.2. Storage Rate According to Hash Code Length of IoT Data Block

The asymmetric storage rate according to the hash code length of the IoT data blocks is evaluated using the rate at which they are stored on the server along the hash code length when composed of IoT data blocks as hash chains. Table 4 compares server storage rates according to hash code paths when IoT data blocks are configured as hash chains. In the results of Table 4, the proposed techniques showed an average 10.3% faster storage rate than conventional techniques. This result is because IoT data are optimized by combining encrypted IoT data into subnets to distribute and store encrypted IoT data in distributed sources using blockchain while configuring it as a building block through polynomial multiplication and security comparison. Furthermore, the proposed technique is due to the classification and storage of IoT data according to hash code length during the load balance process.

#### 4.2.3. IoT Data Storage Efficiency of Servers

The server’s IoT asymmetric data storage efficiency is evaluated using the number of data stored on the server after minimizing errors in IoT data. Table 5 evaluates the efficiency stored on servers after minimizing IoT data errors to ensure the integrity of IoT data by load balancing IoT data. In the result of Table 5, the proposed technique showed an average IoT data storage efficiency of 6.9% higher than conventional techniques because it directly accesses IoT data by weighting IoT data in 64-bit units that are blockchain to reduce bandwidth before forwarding it to the data center. These results show improved storage efficiency because the proposed technique effectively explores the similarity and learning ability of semi-paired data as the IoT data block length increases.

#### 4.2.4. Blockchain Link Creation Time Comparison

Blockchain linkage generation time is evaluated as the generation time of blocks based on probability conditions of linked information propagated along with the network between IoT devices and servers (data centers). Table 6 compares and evaluates the creation time of blockchain links between IoT devices and servers (data centers) of the distributed cloud environment. In the results of Table 6, we obtained an average 17.3% improvement in proposed techniques compared with existing techniques for creating blockchain links in servers (data centers) physically close to IoT devices by linking IoT data to the blockchain. These results are due to IoT devices being physically shorter servers (data centers) located near IoT devices than cloud servers, as well as being able to preprocess and analyze IoT data, resulting in improved IoT data integrity and critical information output.

#### 4.2.5. Integrity Accuracy of IoT Data

The accuracy of IoT data integrity is evaluated using the number of IoT data whose hash results match the entire IoT data transmitted to the server. Table 7 shows the results of analyzing the accuracy of IoT data integrity on the server after delivering IoT data to the cloud server. As with the results of Table 7, the proposed technique obtained an average improvement of 18.3% in the integrity accuracy of IoT data. These result in a lower error rate of IoT data, resulting in improved integrity accuracy of IoT data on the server because that IoT data was weighted to IoT data when configured as blockchain and then synchronized between IoT data using seed values in the hash chain. Furthermore, the proposed technique group managed cumulative use of transactions for processing links between IoT data in blocks of a certain size without directly passing IoT data to cloud servers, indicating higher integrity accuracy of IoT data as data collected from IoT devices increases.

## 5. Conclusions

IoT technology seeks to collect and analyze data generated by IoT devices in cloud environments. In particular, many studies are being conducted socially to minimize network overhead and data processing latency. However, as the cloud environment changes rapidly, IoT devices are required to guarantee the integrity of IoT data for a wide variety of information. In this paper, we propose a technique for optimizing integrity verification of IoT data by combining heterogeneous IoT data into a blockchain in segmentation or cloud environments. The proposed technique maintained the reliability of IoT data after encrypting it using blockchain to record it in a distributed ledger. The proposed technique uses polynomial multiplication and security comparison to allow IoT data to be constructed as building blocks to be optimized for distributed environments. Furthermore, the proposed technique has hash-processed IoT data to minimize errors in their integrity, and then load-balance is performed. Such a process allows minimum errors in collected IoT data while ensuring the linkage of IoT data. As a result of performance evaluation, the proposed techniques reduced IoT data processing time by an average of 2.54 times compared to existing techniques, and blockchain generation time improved by an average of 17.3% when linking IoT data. The asymmetric storage efficiency of IoT data according to hash code length was improved by 6.9% on average over existing techniques. The storage speed of IoT data blocks according to hash code length was shown to be 10.3% faster on average than previous techniques. Integrity accuracy of IoT data was improved by 18.3% on average over existing techniques. V. Yatskive et al. [15], P. Galloet et al. [16] and A. Knischet et al. [7] can cause availability and privacy conservation problems when heterogeneous sources are interconnected, thus ensuring integrity only in private or other optimized blockchains. However, the proposed technique maintains reliability for integrity because it transmits hash block information to blockchain networks so that all IoT devices can trust it. D.Yueet et al. [23] further present a sampling strategy to perform sampling verification more effectively, but the proposed technique generates IoT information type-specific attributes to extract attribute values by weighting IoT data blocks. Based on the results of this study, future studies plan to conduct additional research on complementary aspects of IoT integrity verification by comparing and evaluating the integrity verification error range of IoT information by cloud service.

## Figures and Tables

**Figure 1 sensors-21-04679-f001:**
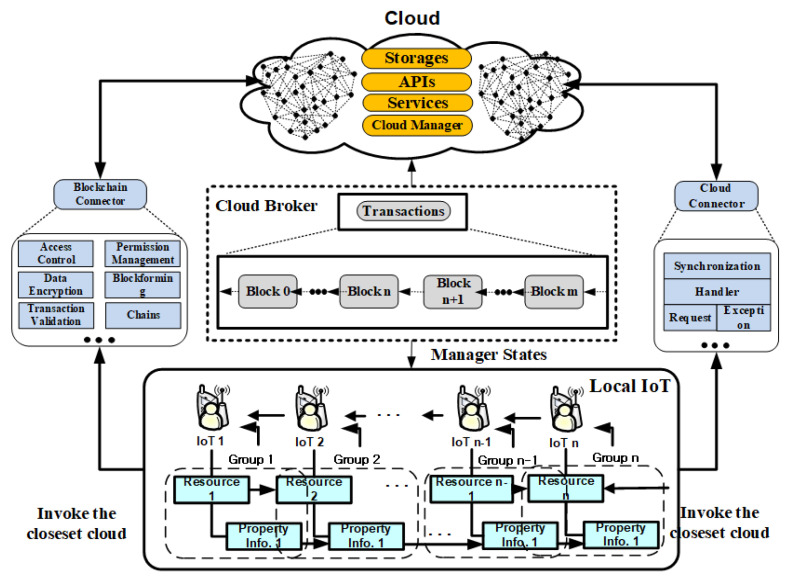
Blockchain-based Internet of Things (IoT) integrity verification data connection structure.

**Figure 2 sensors-21-04679-f002:**
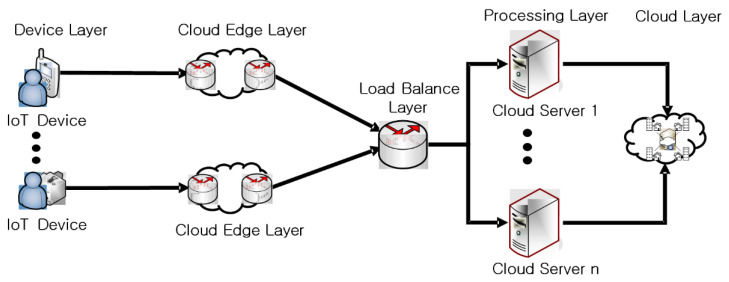
System layer configuration.

**Figure 3 sensors-21-04679-f003:**
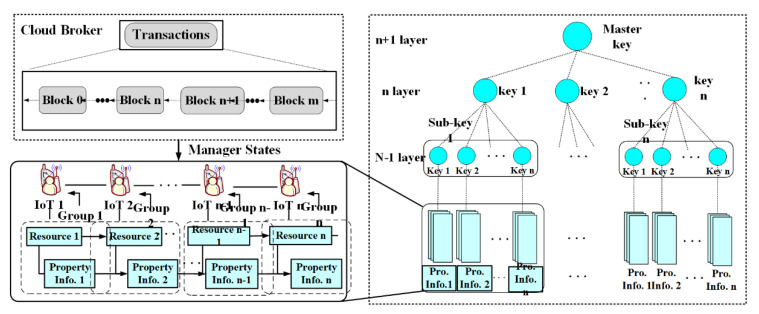
Properties in the creation of blockchain-based IoT data.

**Figure 4 sensors-21-04679-f004:**
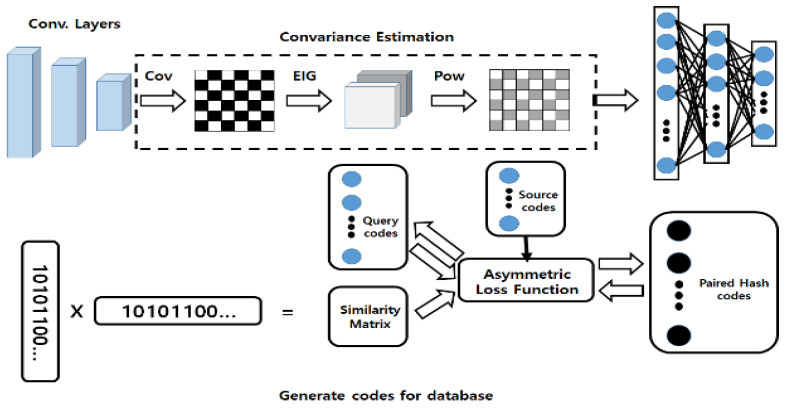
Block Processing for hash processing of IoT data.

**Figure 5 sensors-21-04679-f005:**
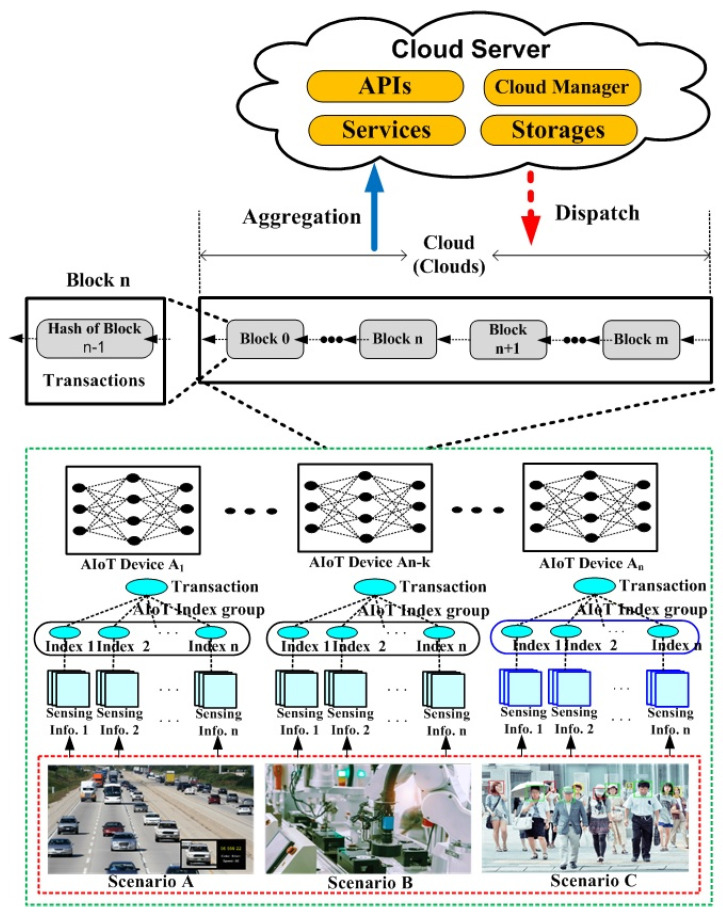
IoT block processing for hash processing.

**Figure 6 sensors-21-04679-f006:**
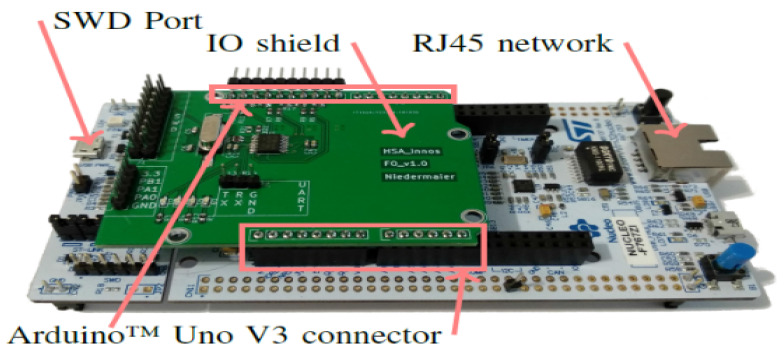
IoT device used in performance evaluation.

**Table 1 sensors-21-04679-t001:** Simulation environment.

Parameter	Value
Dataset Size	30,000
Training Set	26,000
Test Set	4000
CPU (Central Processing Unit)	2.1 Gbps
Memory	128 GB
OS	Ubuntu 16.04.6
Deep Learning Toolbox	MatConvNet

**Table 2 sensors-21-04679-t002:** Environment parameters.

Parameter	Value
CPU	Intel Xeon CPU E5-2630 @ 2.1GHz
GPU (Graphics Processing Units)	2080 Ti
memory	128 G
The transmit/receive the power of the IoT	0.2 W/0.1 W
The network coverage radius	150 m
The static circuit power P	0.02 W
The path loss exponent ε	2
The subnet tree depth	4
The available bandwidth for $\beta_{S}/\beta_{IoT}$	10 MHz/5 MHz
The power of noise $\pi^{2}$	−174 dBm/Hz
Subnet storage capacity	0.5 TB
Input data size $D_{m,\;n}$	2 kbits/s
Delay threshold $\tau_{m,n}$	8 s
Link capacity $L_{m,n}$	10 Gbps
Poisson lambda	85%
Data generation span	10 min
Max access count	50
The unit price of energy $\varphi_{e}$	0.15 Token/J

**Table 3 sensors-21-04679-t003:** Evaluation of IoT data processing time using Raspberry Pi 3.

Units: Process Number/Second										
Division	Serving Thing				Requesting Thing					
Number	1	2	5	10	1	2	5	10	15	20
Not using Blockchain	2.354	3.729	7.685	9.953	1.126	2.78	4.86	5.93	7.26	8.51
Using Blockchain	5.713	6.371	15.75	19.47	3.529	6.08	10.76	14.03	19.76	24.39

**Table 4 sensors-21-04679-t004:** Storage rate according to hash code length of IoT data block.

Units: ms					
Number	Hash Code Length				
	16 bit	32 bit	64 bit	128 bit	256 bit
Not using Blockchain	0.418	0.561	0.754	0.842	0.883
Using Blockchain	0.284	0.372	0.521	0.638	0.691

**Table 5 sensors-21-04679-t005:** Storage rate according to hash code length of IoT data block.

Units: %					
Number	Block Size of IoT Data				
	16 bit	32 bit	64 bit	128 bit	256 bit
Not using Blockchain	63.741	66.093	70.634	76.507	80.025
Using Blockchain	67.307	71.628	76.032	82.354	84.659

**Table 6 sensors-21-04679-t006:** Blockchain link creation time comparison.

Units: ms																
Number	Number of Sever (Datacenter)															
	1				2				5				10			
	Number of IoT Sensor				Number of IoT Sensor				Number of IoT Sensor				Number of IoT Sensor			
	5	10	25	50	5	10	25	50	5	10	25	50	5	10	25	50
Not using Blockchain	2.451	3.768	5.398	8.459	1.895	2.121	3.214	5.031	0.942	1.137	1.612	2.251	0.892	0.908	1.021	1.241
Using Blockchain	1.624	2.784	4.754	6.845	1.211	1.705	2.347	3.652	0.702	0.932	1.024	1.652	0.452	0.589	0.801	1.027

**Table 7 sensors-21-04679-t007:** Comparison of integrity accuracy of IoT data.

Units: %					
Number	Number of IoT Sensor				
	10	25	50	100	200
Not Using Blockchain	80.241	77.204	73.214	71.032	69.354
Using Blockchain	84.147	80.0357	78.324	74.3652	72.325

## Data Availability

Publicly available datasets were analyzed in this study. The data presented in this study are available on request from the corresponding author.

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
