# Peer review of "Blockchain Processing Technique Based on Multiple Hash Chains for Minimizing Integrity Errors of IoT Data in Cloud Environments"

_sensors, 2021, doi:10.3390/s21144679_

Round 1
Reviewer 1 Report
Title of Paper: - Blockchain Processing Technique based on Multiple Hash Chains for minimize integrity errors of IoT data in Cloud Environments.
Detail Comments: -
The authors in this article propose an optimized hash processing technique that can minimize
the integrity errors of data generated by numerous IoT devices deployed in distributed cloud environments. This is an interesting and insightful study, that contributes towards further creation of new businesses through the development of applications that can be used in cloud environments and widespread adoption of IoT smart devices.
The proposed technique aims to maintain reliability among devices supplying IoT data by recording encrypted IoT data in a distributed ledger using blockchain. Blockchain can enrich the IoT by providing a trusted sharing service, where information is reliable and can be traceable. The author performs two-way authentication and distributed processing by separating the index information of IoT data located at different layers by n-bit size for integrity verification of IoT data bound by blockchain, ensuring the reliability of IoT data.
The critical components involved in the design of this method are:
- Optimizing integrity verification of IoT data by grouping encrypted IoT data into subnets and storing data in distributed sources using blockchain.
- Use polynomial multiplication and security comparisons to ensure illegal intervention by third parties.
- Optimize IoT data through load balance to secure the linkages of IoT data and the speed of sending and receiving.
The abstract of the paper is brief enough to indicate the purpose and significance of the work. The originality of this paper is sound, and the readability and structure are well presented. The study is conducted in both experimental and theoretical point of view.
Overall, this paper can be accepted because the author proposes novel idea for blockchain-based IoT data processing studies that are continuously trying to minimize network delays in cloud environments, can minimize bottlenecks in overlay networks and latency in processing IoT data. There are some grammatical mistakes and spelling errors need to be corrected. I also recommend the following correction and/ or inclusions:
- The title of the paper should be “Blockchain Processing Technique based on Multiple Hash Chains to minimize integrity errors of IoT data in Cloud Environments”
Or “Blockchain Processing Technique based on Multiple Hash Chains for minimizing integrity errors of IoT data in Cloud Environments”
- In section 3.2, on page 5, need to correct spelling. It will be “Load balance layer”
- In section 4.1 on page 10 need to correct spelling. It will be “…. 100-200B and the threshold …”.
Author Response
Reviewer #1:
Detail Comments: -
The authors in this article propose an optimized hash processing technique that can minimize
the integrity errors of data generated by numerous IoT devices deployed in distributed cloud environments. This is an interesting and insightful study, that contributes towards further creation of new businesses through the development of applications that can be used in cloud environments and widespread adoption of IoT smart devices.
The proposed technique aims to maintain reliability among devices supplying IoT data by recording encrypted IoT data in a distributed ledger using blockchain. Blockchain can enrich the IoT by providing a trusted sharing service, where information is reliable and can be traceable. The author performs two-way authentication and distributed processing by separating the index information of IoT data located at different layers by n-bit size for integrity verification of IoT data bound by blockchain, ensuring the reliability of IoT data.
The critical components involved in the design of this method are:
Optimizing integrity verification of IoT data by grouping encrypted IoT data into subnets and storing data in distributed sources using blockchain.
Use polynomial multiplication and security comparisons to ensure illegal intervention by third parties.
Optimize IoT data through load balance to secure the linkages of IoT data and the speed of sending and receiving.
The abstract of the paper is brief enough to indicate the purpose and significance of the work. The originality of this paper is sound, and the readability and structure are well presented. The study is conducted in both experimental and theoretical point of view.
Overall, this paper can be accepted because the author proposes novel idea for blockchain-based IoT data processing studies that are continuously trying to minimize network delays in cloud environments, can minimize bottlenecks in overlay networks and latency in processing IoT data. There are some grammatical mistakes and spelling errors need to be corrected. I also recommend the following correction and/ or inclusions:
The title of the paper should be “Blockchain Processing Technique based on Multiple Hash Chains to minimize integrity errors of IoT data in Cloud Environments”
Or “Blockchain Processing Technique based on Multiple Hash Chains for minimizing integrity errors of IoT data in Cloud Environments”
* Response of Reviewer
As request reviewer,
- I changed the title of the paper as “Blockchain Processing Technique based on Multiple Hash Chains for minimizing integrity errors of IoT data in Cloud Environments”
In section 3.2, on page 5, need to correct spelling. It will be “Load balance layer”
* Response of Reviewer
As request reviewer,
- I corrected spelling as load balance layer.
In section 4.1 on page 10 need to correct spelling. It will be “…. 100-200B and the threshold …”.
* Response of Reviewer
As request reviewer,
- I corrected spelling in section 4.1.
“… 100200B and th threshold …“ ““…. 100-200B and the threshold … ” .
Reviewer 2 Report
The topic discussed in the article is relevant in the IoT subject, nevertheless, several improvements need to be taken into account before the paper publication.
a) In the introduction section, it would be appropriate to describe more clearly the strategy adopted to evaluate each research goal identified.
b) Regarding the related works section, instead of enumerating a list of papers, it is necessary to analyze their relationship and limitations in relation with the research presented from the global to the particular point of view.
c) In the evaluation section, it is necessary to specify clearly how each of the elements described in the systems architecture and IoT Data Block processing have been implemented for evaluation. It is not clear how the elements described in section 3 have been considered in the evaluation phase. One of the research goals indicate the minimization of the integrity errors of data generated by numerous IoT devices deployed in distributed cloud environments. It is also unclear how the numerous IoT devices are represented in the evaluation environment.
d) The relationship among the research goals (to guarantee the integrity of IoT data by hashing IoT data from servers adjacent to IoT devices to blockchain instead of cloud servers) and the evaluation approach (evaluation of data processing time, data storage efficiency and accuracy of IoT data) must be presented in a more clear and reasoned way.
e) Even more, in evaluation section, it is necessary to specify clearly the tests proposed and carried out to evaluate all the items considered in evaluation: data processing time, data storage efficiency and accuracy of IoT data .
f) In the conclusion section, it is necessary to discuss more deeply the comparison of your results with other relevant research works using some of the references included in the Related Works section.
Author Response
Reviewer #2:
Comments and Suggestions for Authors
The topic discussed in the article is relevant in the IoT subject, nevertheless, several improvements need to be taken into account before the paper publication.
- a) In the introduction section, it would be appropriate to describe more clearly the strategy adopted to evaluate each research goal identified.
* Response of Reviewer
As request reviewer,
- In the introduction section, I corrected for describe more clearly the strategy adopted to evaluate each research goal identified.
In this paper, we propose an optimized hash processing technique that enables hierarchical distributed processing with an n-bit-size blockchain to minimize the loss of data generated from IoT devices deployed in distributed cloud environments. The proposed technique minimizes IoT data integrity errors as well as strengthening the role of intermediate media acting as gateways by interactively authenticating blockchains of n bits into n+1 and n-1 layers to normally validate IoT data sent and received from IoT data integrity errors. In particular, the proposed technique ensures the reliability of IoT, information by validating hash values of IoT data in the process of storing index information of IoT data distributed in different locations in blockchain to maintain the integrity of the data. Furthermore, the proposed technique ensures the linkage of IoT data by allowing minimal errors in the collected IoT data while simultaneously grouping the linkage information of IoT data, thus optimizing the load balance after hash processing the IoT data.
- b) Regarding the related works section, instead of enumerating a list of papers, it is necessary to analyze their relationship and limitations in relation with the research presented from the global to the particular point of view.
* Response of Reviewer
As request reviewer,
- I added the limitations of the relevant studies to the related works section.
- c) In the evaluation section, it is necessary to specify clearly how each of the elements described in the systems architecture and IoT Data Block processing have been implemented for evaluation. It is not clear how the elements described in section 3 have been considered in the evaluation phase. One of the research goals indicate the minimization of the integrity errors of data generated by numerous IoT devices deployed in distributed cloud environments. It is also unclear how the numerous IoT devices are represented in the evaluation environment.
* Response of Reviewer
As request reviewer,
- I added to section 4 about how the factors described in section 3 were considered in the evaluation phase.
In simulations, IoT devices that makeup distributed networks exploit IoT Arduino equipment. IoT devices randomly generate packets and send them to other IoT devices, and each AP's base station consists of a multi-mesh network. Each IoT device that makes up the network is identified by an IoT recognizer. Furthermore, each base station allows the server to collect and analyze IoT information transmitted and received from the antenna in real-time. At this time, we assume that all IoT information packets are passed on to the relative IoT device in time and that all IoT devices can be synchronized.
- d) The relationship among the research goals (to guarantee the integrity of IoT data by hashing IoT data from servers adjacent to IoT devices to blockchain instead of cloud servers) and the evaluation approach (evaluation of data processing time, data storage efficiency and accuracy of IoT data) must be presented in a more clear and reasoned way.
* Response of Reviewer
As request reviewer,
- In order to clearly indicate the research goal and the proposed method, the following contents were added to Section 4.1.
In the proposed technique, IoT data from servers adjacent to IoT devices are hashed into the blockchain rather than cloud servers to ensure the integrity of IoT data. Each IoT device participating in the blockchain operates according to the generated events (e.g., block generation and message exchange). Each event uses parameters such as block size, number of nodes, number of adjacent nodes of all nodes, block generation capacity of all nodes, network bandwidth between each pair of regions (up/down), and average network delay between each pair of regions. Each block is generated by probability conditions according to the importance of linked information and propagates along with blockchain networks, evaluating data processing time, data storage efficiency, and IoT data accuracy.
- e) Even more, in evaluation section, it is necessary to specify clearly the tests proposed and carried out to evaluate all the items considered in evaluation: data processing time, data storage efficiency and accuracy of IoT data .
* Response of Reviewer
As request reviewer,
- I added all the items that are considered in evaluation (data processing time, data storage efficiency, IoT data accuracy, etc.).
In 4.2.1,
IoT data processing time is evaluated using the number of processed transactions per second that match the hash value among all transactions entered into the server (data center). Data processing time utilized the total number of blocks entered into the server and the number of blocks processed per second.
In 4.2.2,
The asymmetric storage rate according to the hash code length of IoT data blocks is evaluated using the rate at which they are stored on the server along the hash code length when composed of IoT data blocks as hash chains.
In 4.2.3,
The server's IoT asymmetric data storage efficiency is evaluated using the number of data stored on the server after minimizing errors in IoT data.
In 4.2.4,
Blockchain linkage generation time is evaluated as the generation time of blocks based on probability conditions of linked information propagated along the network between IoT devices and servers (data centers).
In 4.2.5
The accuracy of IoT data integrity is evaluated using the number of IoT data whose hash results match among the entire IoT data transmitted to the server.
- f) In the conclusion section, it is necessary to discuss more deeply the comparison of your results with other relevant research works using some of the references included in the Related Works section.
* Response of Reviewer
As request reviewer,
- In the conclusion section, I added comparing the proposed technique to other relevant research works.
- Yatskive et. al [15], P. Galloet et. al [16] and A. Knischet et. al [7] can cause availability and privacy conservation problems when heterogeneous sources are interconnected, thus ensuring integrity only in private or other optimized blockchains. However, the proposed technique maintains reliability for integrity because it transmits hash block information to blockchain networks so that all IoT devices can trust it. And D.Yueet et. Al [23] further presents a sampling strategy to perform sampling verification more effectively, but the proposed technique generates IoT information type-specific attributes to extract attribute values by weighting IoT data blocks.
Reviewer 3 Report
Title should be rectified in terms of grammar.
There’s repetition of same information in multiple paragraphs like “IoT devices have been used in various fields”. Please correct this, it occurred in 3 different paragraphs.
Also, the information for proposed technique is repeated multiple times, please use this only where applicable but in a succinct manner.
A brief explanation regarding block chain technique and why do we need it for IoT is required.
The quality of Figure 3 should be improved.
Why Arduino Uno is used, why not Raspberry Pi?
Author Response
Reviewer #3:
Comments and Suggestions for Authors
Title should be rectified in terms of grammar.
* Response of Reviewer
As request reviewer,
- I changed the title of the paper as “Blockchain Processing Technique based on Multiple Hash Chains for minimizing integrity errors of IoT data in Cloud Environments”
There’s repetition of same information in multiple paragraphs like “IoT devices have been used in various fields”. Please correct this, it occurred in 3 different paragraphs.
* Response of Reviewer
As request reviewer,
- I modified the statements such as "IoT devices have been used in variable fields" so that they are not used equally in multiple paragraphs.
Also, the information for proposed technique is repeated multiple times, please use this only where applicable but in a succinct manner.
* Response of Reviewer
As request reviewer,
- Information about the proposed technology has been modified to be concise only where applicable, so that it does not repeat itself many times.
A brief explanation regarding block chain technique and why do we need it for IoT is required.
The quality of Figure 3 should be improved.
* Response of Reviewer
As request reviewer,
- We improved the quality of Figure 3.
Why Arduino Uno is used, why not Raspberry Pi?
* Response of Reviewer
As request reviewer,
- In the proposed technique, Arduino Uno was used to perform the role of simply generating information generated by IoT devices and passing it to gateways or servers (data centers). On the other hand, Raspberry Pi was used as a device that acted as a gateway in the network. In particular, Raspberry Pi has been used as a role in reducing network latency and processing time by giving the ability to preprocess and analyze data transmitted from IoT devices, allowing only preprocessed data to be delivered to the server.
Reviewer 4 Report
"Blockchain Processing Technique based on Multiple Hash Chains for minimize integrity errors of IoT data in Cloud Environments" is a very interesting paper with possible future practically implementations.
Is interesting to validate the proposed solutions on a real process, implemented on Real Time hardware (during real noise presence), because the proposed solution is validated only on/in simulation.
Author Response
Reviewer #4:
Comments and Suggestions for Authors
"Blockchain Processing Technique based on Multiple Hash Chains for minimize integrity errors of IoT data in Cloud Environments" is a very interesting paper with possible future practically implementations.
Is interesting to validate the proposed solutions on a real process, implemented on Real Time hardware (during real noise presence), because the proposed solution is validated only on/in simulation.
Some (possible) short observations:
- please provide more details about possible case study;
* Response of Reviewer
As request reviewer,
- I added more details about the case study in section 4.
- in figure 5 please describe what represent scenario A, B, C;
* Response of Reviewer
As request reviewer,
- Scenarios A, B, and C in Figure 5 represent the environment in which IoT information is collected. I added the information about this to Section 3.6.
In Figure 5, IoT data is collected at regular time intervals depending on the speed and presence, and absence of the nodes. The scenarios used in Figure 5 are distinguished by the speed of movement of the nodes and their presence and absence. Scenario A refers to an environment that travels at high speed (more than 30 km/h), Scenario B refers to an environment that collects IoT information in a fixed place such as an industrial IoT, and Scenario C refers to an environment that collects information on IoT devices that travel at low speed (less than 30 km/h).